# Translation, validation and cultural adaptation of the Arabic version of the HIV knowledge questionnaire (HIV-Kq-18)

**Mohamed Terra**[1,2]*, **Mohamed Baklola**[1,2]*, **Elfatih A. Hasabo**[3]*, **Dina Gamal Shaheen**[4], **Abdel-Hady El-Gilany**[4], **ARO team of collaborators**[¶]

**1** Faculty of Medicine, Mansoura University, Mansoura, Egypt, **2** Alpha Research Organization (ARO), Mansoura, Egypt, **3** Faculty of Medicine, University of Khartoum, Khartoum, Sudan, **4** Faculty of Medicine, Public Health and Community Medicine Department, Mansoura University, Mansoura, Egypt

¶ Membership list can be found in the Acknowledgments section

* Mohamedterra75@std.mans.edu.eg (MT); Mohamedbaklola@std.mans.edu.eg (MB); elfatih.ahmed.hasabo@gmail.com (EAH)

## Abstract

### Background

Although the number of new HIV infections is declining in most regions of the world, the Middle East is one of the regions with a rapidly growing HIV epidemic, with Egypt having the fastest-growing epidemic, with a 76 percent increase in the number of cases. One of the major factors contributing to this trend is the general public's lack of knowledge about the disease. The HIV Knowledge Questionnaire-18 (HIV-KQ-18) is one of the most widely used instruments for assessing HIV/AIDS knowledge and has been translated into several languages. This study examined the validity of the Arabic version of the HIV-18-KQ as well as its adaptation among Arab undergraduates.

### Methods

The HIV-18-KQ was both forward and back-translated. The translation was reviewed by an expert committee of eight experts. The final version was created and distributed to undergraduates from five Arabic countries: Egypt, Sudan, Yemen, Jordan, and Algeria. The validity of the Arabic version of the HIV-18-KQ was evaluated using internal consistency and construct validity. Internal consistency was tested using the Kuder-Richardson formula 20 (KR-20), and construct validity was evaluated using an exploratory factor analysis with a polychoric correlation matrix.

### Results

The majority of the translated items were easy to understand. The Arabic HIV-18-KQ was deemed culturally appropriate by the expert committee. This study included 1745 university students, including 956 (54.5%) males and 798 (45.5%) females, with 33.4% from Egypt. Based on the acceleration factor approach to interpreting the scree plot in the factor analysis, it was preferable to use only one factor, which is consistent with the original version of HIV-45-KQ. The KR-20 value was 0.73, indicating good internal reliability.

**Data Availability Statement:** The datasets used and/or analysed during the current study are

available from the following DOI link: https://doi.org/10.6084/m9.figshare.22236550.v1.

**Funding:** The author(s) received no specific funding for this work.

**Competing interests:** The authors have declared that no competing interests exist.

## Conclusion

This study demonstrates that the Arabic version of the HIV-18-KQ is a valid and reliable tool for assessing HIV-related knowledge in Arabic-speaking countries.

## Introduction

The Human Immunodeficiency Virus (HIV) and acquired immunodeficiency syndrome (AIDS) continue to be major worldwide health concerns, with intense international and local efforts to organize resources to combat the epidemic [1]. According to the Joint United Nations Programme on HIV/AIDS (UNAIDS), there would be 38.4 million HIV-positive people globally in 2021, with 650 thousand dying of AIDS-related illnesses. Furthermore, it is estimated that one in six people with HIV (15%) are still unaware they are infected [2]. The Sustainable Development Goals (SDGs) sparked a coordinated global effort to combat HIV/ growing AIDS prevalence [3,4]. To combat the increased prevalence of HIV, more proactive initiatives with preventative education are urgently needed, and the global community must evaluate the SDGs' progress throughout all globe areas [3].

Although the HIV pandemic is witnessing a decline in the number of new infections in most regions of the world, the Middle East and North Africa Region (MENA) is one of the regions of the world with a rapidly increasing HIV epidemic [5]. Egypt has a relatively low estimated number of people living with HIV, which was approximately 11,000 by the end of 2016, relative to the total population. However, the country is experiencing the fastest-growing epidemic in the Middle East and North Africa Region (MENA), with a 76% increase in the number of cases between 2010 and 2016 [6]. This alarming trend is further highlighted by the annual increase in the number of newly confirmed cases, which is between 25–30%. These worrying signs indicate a dire need for increased investments to prevent further epidemic growth and avoid the failure of controlling the epidemic [6]. Behavioral change provides the ultimate and cheapest protection against HIV infection, notwithstanding good advances in HIV therapy, because there is no cure or vaccine for HIV, and people with less HIV knowledge are more likely to participate in risky behaviors [7]. Unfortunately, data on the population's level of knowledge of HIV and trends in the region is poor, under-reporting is likely, and exact figures or the specific causes of them are challenging to obtain [5].

A lack of Arabic-language tools for testing and measuring HIV knowledge could explain this under-reporting in describing knowledge levels. The HIV Knowledge Questionnaire (HIV-KQ-18) instrument has been shown to be a valid, reliable (Cronbach's alpha at 0.75–0.89), stable, sensitive, and appropriate instrument for all people, including those with low literacy levels [8]. HIV-KQ-18 is available in a wide range of languages and has been used in a variety of contexts, including Spanish, Greek, and Indonesian [9,10]. There have been no studies that thoroughly adapt and validate this instrument for the Arab adult population. As a result, the objective of this study was to translate, validate, and adapt the HIV-KQ-18 instrument on university students in our MENA region.

## Methods

### Sampling and procedures

Participants were university students between the age group of 18 and 25. They were from five Arab nations; Algeria, Egypt, Sudan, Yemen, and Jordan. These five countries represent a diverse range of cultures, traditions, and beliefs throughout the Middle East and North Africa.

In each country, a team of data collectors was assembled. This group received standardized instruction on how to approach Students online. From May 2022 to June 2022, each collaborator oversaw the collection of responses by posting the questionnaire on official student social media groups from all academic years in order to reach all students.

Participants completed an anonymous online questionnaire that was processed using Google Form to guarantee a broad reach and ease of use. Participants chose to take part willingly and without remuneration.

There are no precise standards for the sample size required to validate or study tools due to the wide variety of tools and the number of items they contain. The rule of thumb has been at least 10 participants for each scale item, implying that a 10.1 ratio of respondents to items is ideal [11]. Because the instrument has 18 items, the minimum sample size required is 180 students, which is then multiplied by 5 for the design effect, for a total sample of 900. Since larger samples are usually preferable to smaller ones, when conducting exploratory factor analysis, a total sample of 1745 was collected [12].

## Instrument

The instrument contained the HIV-KQ-18 [10] and sociodemographic variables. HIV-KQ18 is a shortened form of HIV-KQ-45. Professor Michael P. Carey, Ph.D. granted permission to use and translate the HIV-KQ-18 instrument (Director of Miriam Hospital's Center for Behavioral and Preventive Medicine). The HIV-KQ-18 instrument is more oriented toward HIV/AIDS infection and transmission prevention. Each of the 18 items on this instrument has three options: "true," "false," or "don't know." Five of the items are true (items number: 1, 4, 11, 14, and 17), whereas the other 13 are false. The correct response receives a 1, whereas incorrect or "don't know" answers receive a 0.

The collected sociodemographic data included sex, age, marital status, nationality, residence, and field of study. Participants were divided into two groups according to whether they have a medical educational background (Medicine, Pharmacy, Nursing, or Dentistry) or not.

They were informed that the data submitted from the questionnaire would be anonymous, confidential, and used only for research purposes in order to protect confidentiality. Also, access to the dataset was restricted to researchers.

## Arabic translation of HIV-KQ 18

Two independent linguists translated the English version into Arabic, and the two versions were compared to come to an agreed-upon starting version. From a linguistic standpoint, this first version was revised (S1 Appendix).

To confirm the correctness of the translation and the consistency of the synonyms, the Arabic version was retranslated into English by two additional linguists who were ignorant of the original English version.

## Questionnaire validity

A jury of eight experts in public health and microbiology assessed the content validity. The Arabic version of the scale was evaluated for clarity, relevance, and content translation. The experts were asked to review each item separately using a three-point ordinal scale regarding clarity and relevance while translation using correct or incorrect. They were asked to give suggestions if they found them incorrect. The content validity index (CVI) was determined at three levels: item (I-CVI), expert (E-CVI), and scale (S-CVI) [13]. To calculate the CVI at the item level (I-CVI), divide the number of experts who rated the item as relevant or clear (rating 3) by the total number of experts. The CVI per expert (E-CVI) was determined by dividing the

number of items rated by three. The item was appropriate if the CVI was greater than 0.79. It needs to be revised if it was between 0.70 and 0.79. It was deleted if it was less than 0.70. The CVI for the full scale (SCVI) was calculated using the S-CVI, by adding all I-CVI for relevancy divided by the number of items. If S-CVI was more than or equal to 0.90, the scale was considered valid as a tool [14]. An overview of the entire study process is shown in Fig 1.

### Reliability and factor analysis

Internal consistency was tested using the Kuder-Richardson formula 20 (KR-20) as all items employed a dichotomous response [15]. KR-20 score above 0.70 is generally considered to represent a reasonable level of internal consistency reliability and indicates strong item homogeneity.

Due to the dichotomous nature of HIV-KQ-18, an exploratory factor analysis (EFA) with a polychoric correlation matrix was used to examine construct validity [16,17].

The number of factors that might be retained was determined using eigenvalues, parallel analysis, optimal coordinates, and acceleration factor to eliminate subjectivity in reading the scree plot. If these parameters provided opposing recommendations for the number of factors to retain, other considerations were used, such as the feasibility of interpreting underlying factors [13].

All statistical analyses were carried out with SPSS version 25 (IBM Corp., Armonk, New York, USA), R version 4.1.0, and R Studio Version 1.4.1717 using the polycor, n Factors, validate R, and EFA.dimensions packages. The statistical significance level was set at P-value < 0.05.

### Ethics approval and consent to participate

The study proposal was approved by the Institutional Review Board (IRB) of Mansoura University's Faculty of Medicine, with the proposal code R.22.04.1683.R1. Google Form was used to conduct an anonymous online questionnaire submitted by participants. Participants participated voluntarily and without compensation. All methods were carried out in accordance with the relevant ethical guidelines and regulations. On the first page of the online form, participants were informed that by filling the questionnaire, they were giving us permission to use the data for research purposes (Informed Consent) and that they had the option to stop filling out the form at any time. So, after being told what the questionnaire was for, all of the participants who filled it out granted their permission.

## Results

### Participant characteristics

There were 1754 participants from five Arab countries. All the participants were between the ages of 17 and 25, with the majority being male (54.5%), single (83.5%), Egyptian (33.4%), urban (80.7%), and studying health-related sciences (57.6%). Table 1 contains detailed socio-demographic information about the participants.

### Validity

Table 2 shows that the I-CVI ranged from 0.91 to 1.0 in different items of both relevance and clarity. While the E-CVI ranged from 0.92 to 1.0 in different items of both relevance and clarity. The S-CVIs were 0.98 for relevance and clarity.

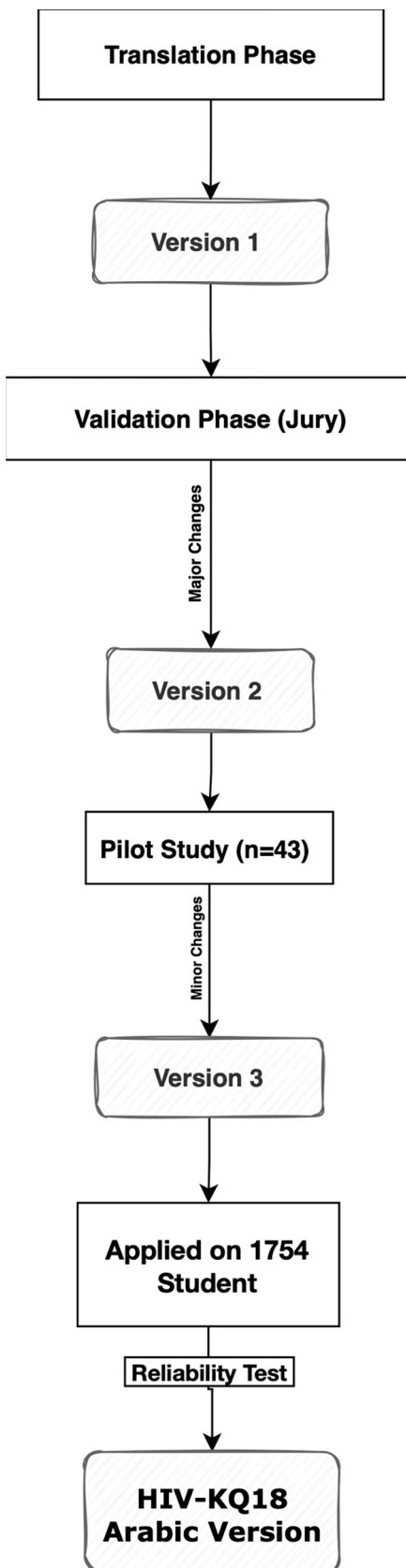

**Fig 1. Adaptation and validation of HIV-KQ-18 for Arabic population.**

**Table 1. Participants' characteristics.**

| Variables | n | Percent |
|---|---|---|
| **Age** | | |
| 17–20 | 302 | 17.2 |
| 21–23 | 737 | 42 |
| 24–25 | 715 | 40.8 |
| **Sex** | | |
| Female | 798 | 45.5 |
| Male | 956 | 54.5 |
| **Martial Status** | | |
| Single | 1465 | 83.5 |
| Engaged | 97 | 5.5 |
| Married | 192 | 10.9 |
| **Nationlity** | | |
| Algeria | 339 | 19.3 |
| Egypt | 585 | 33.4 |
| Sudan | 414 | 23.6 |
| Jordan | 209 | 11.9 |
| Yemen | 207 | 11.8 |
| **Field of Study** | | |
| Non-Medical Education | 744 | 42.4 |
| Medical Education | 1010 | 57.6 |
| **Residence** | | |
| Rural | 338 | 19.3 |
| Urban | 1416 | 80.7 |
| **Total Participants** | **1754** | **100** |

## Cultural adaptation

Following the pilot study and jury, we conducted rounds of evaluation for content relevance and cultural background debugging. It is recommended to replace the word "Penis" with the word "Male genital organ" which is a less intense and culturally appropriate word in Arab countries.

Recommendations from the pilot study indicated that most people are unfamiliar with the HIV virus but are familiar with AIDS; as a result, we made a point to emphasize that HIV is the cause of AIDs, which is widely known among Arabs.

## Item analysis

The percentage of correct answers and the Corrected Item-Total Correlation are shown in Table 3. In terms of difficulty, questions 12 and 15 were the most difficult, while questions 14 and 1 were the easiest. Most participants correctly answered Question 14 (94.8 percent). However, item 12 proved to be the most difficult, with less than 30% of participants correctly answering it (15.2 percent).

There was a range of 30 to 80 in the percentage of correct answers for the other 16 items, so they were acceptable. Correlations were less than 0.3 for eight items (numbers 1, 4, 6, 9, 11, 12, 14, and 17), while only five items (numbers 6, 9, 12, 14, and 17) were less than 0.25 in their corrected item totals. Item 17 has the fewest correlations.

**Table 2. Content validity indices.**

| Item | I-CVI for relevance | I-CVI for clarity | Expert | E-CVI for relevance | E-CVI for clearance |
|---|---|---|---|---|---|
| 1 | 1 | 1 | 1 | 1 | 1 |
| 2 | 1 | 1 | 2 | 1 | 1 |
| 3 | 1 | 0.91 | 3 | 1 | 1 |
| 4 | 0.96 | 1 | 4 | 1 | 1 |
| 5 | 0.92 | 0.96 | 5 | 0.92 | 0.96 |
| 6 | 0.92 | 0.96 | 6 | 0.92 | 0.94 |
| 7 | 1 | 1 | 7 | 1 | 0.96 |
| 8 | 1 | 0.92 | 8 | 1 | 1 |
| 9 | 0.92 | 1 | | | |
| 10 | 1 | 1 | | | |
| 11 | 1 | 1 | | | |
| 12 | 1 | 1 | | | |
| 13 | 0.96 | 1 | | | |
| 14 | 1 | 1 | | | |
| 15 | 1 | 1 | | | |
| 16 | 1 | 0.96 | | | |
| 17 | 1 | 1 | | | |
| 18 | 1 | 1 | | | |
| **S-CVI/Ave** | **0.98** | **0.98** | | | |
| **S-CVI/UA** | **0.72** | **0.77** | | | |

## Factor analysis

Only five factors could be retained in the EFA based on eigenvalues, one factor based on parallel analysis, optimal coordinates, and the acceleration factor as shown in Fig 2. There were some difficulties in interpreting five factors on eigenvalues because the cross-loading of some

**Table 3. Item analysis of HIV-KQ-18 Arabic version.**

| Item | Percentage of Correct Answers | Corrected Item-Total Correlation |
|---|---|---|
| Q1 | 77.10% | 0.249 |
| Q2 | 69.70% | 0.336 |
| Q3 | 51.70% | 0.334 |
| Q4 | 69.00% | 0.284 |
| Q5 | 61.60% | 0.397 |
| Q6 | 45.30% | 0.243 |
| Q7 | 56.60% | 0.356 |
| Q8 | 62.30% | 0.391 |
| Q9 | 37.90% | 0.233 |
| Q10 | 62.20% | 0.45 |
| Q11 | 40.40% | 0.255 |
| Q12 | **15.20%** | 0.225 |
| Q13 | 73.70% | 0.467 |
| Q14 | **94.80%** | **0.18** |
| Q15 | 31.90% | 0.301 |
| Q16 | 51.10% | 0.389 |
| Q17 | 49.90% | **0.102** |
| Q18 | 36.70% | 0.372 |

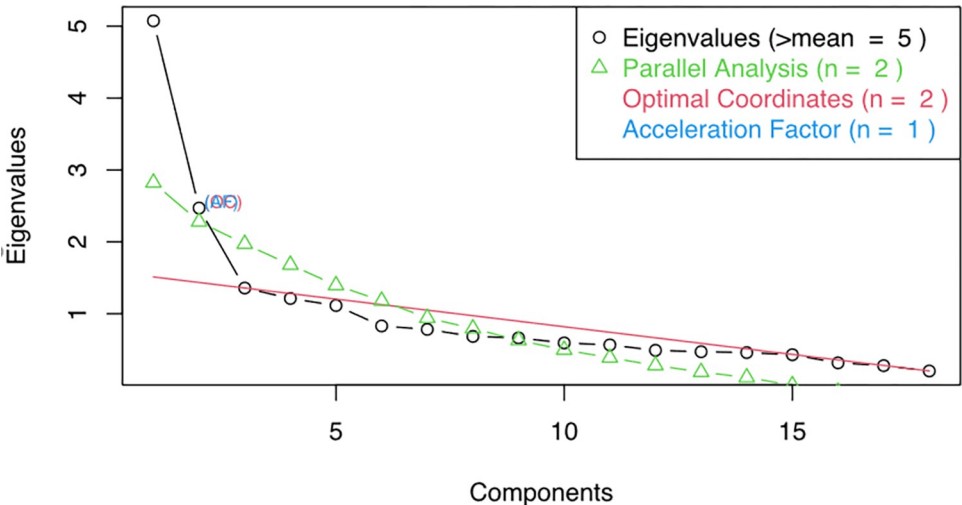

**Fig 2. The non-graphical solution of Scree plot to determine the number of factors to be retained.**

items made it difficult. As a result, one factor with 17 items was retained from the EFA with an RMSEA value of 0.135. There were no items with loading factors lower than 0.30. Reliability is indicated by the KR-20 Coefficient of 0.73 for this factor. The details of the EFA are presented in Table 4.

## Discussion

Currently, there is no validated Arabic tool for assessing the general public's knowledge of HIV. Many studies have been conducted in Arabic countries to assess the knowledge about HIV, but most of them either used the English version of HIV-18 KQ [18] or developed an

**Table 4. Factor analysis of HIV-KQ-18 Arabic version.**

| Item | Factor loading | KR-20 Coefficient |
|------|----------------|-------------------|
| **Q1** | 0.41 | **0.733** |
| **Q2** | 0.52 | |
| **Q3** | 0.47 | |
| **Q4** | 0.4 | |
| **Q5** | 0.59 | |
| **Q6** | 0.35 | |
| **Q7** | 0.51 | |
| **Q8** | 0.6 | |
| **Q9** | 0.39 | |
| **Q10** | 0.66 | |
| **Q11** | 0.36 | |
| **Q12** | 0.41 | |
| **Q13** | 0.74 | |
| **Q14** | 0.41 | |
| **Q15** | 0.47 | |
| **Q16** | 0.57 | |
| **Q17** | **0.3** | |
| **Q18** | 0.55 | |

Arabic tool to assess the knowledge, but their tool wasn't tested on a large scale to assess its validity and reliability [19]. As a result, ours is the first study to translate, validate, and adapt the HIV-18 KQ in the Arabic community.

Content validity can be tested in a variety of ways, one of them is Content Validity Index (CVI) which we used to quantify content validity and it is also the most commonly used method.

Since an average score of Item-level CVI (I-CVI) can be skewed by outliers, therefore this paper looked at the I-CVI, expert-level CVI (E-CVI), and scale CVI (S-CVI) [20].

The number of experts (n = 8) was deemed sufficient for content validation [21]. A minimum I-CVI score of 0.78 is regarded as excellent. Individual items were determined to be important and relevant for measuring content validity. Any S-CVI value between 0.80 and 0.90 is regarded as minimally acceptable [20,21]. In the current study, the relevance and clarity scores for I-CVI, E-CVI, and S-CVI ranged between 0.9 and 1.0.

Our findings suggest that the HIV-KQ-18 Arabic Version is a reliable and valid instrument for use in various Arabic-speaking countries. The adaptation phase of the instrument began with a pilot test, which revealed that all items were clear to participants and there was no difficulty comprehending the translated items. Then we distributed the instrument to our participants. As a result, we extended the instrument's usage to general undergraduates versus prior studies that mainly utilized the tool in its English version on Arab medical students [18].

The number of factors that should be maintained in order to ensure construct validity varies depending on the specifics of the situation. We chose to keep one factor based on the parallel analysis and acceleration factor from the scree plot, which is consistent with the original HIVKQ-45, which suggested only one factor labeled HIV knowledge [22]. Although additional studies are required, we believe the structural validity of the HIV-KQ18 is acceptable. The corrected item-total correlation indicates that the correlation was less than 0.2 for question 14 and 17, which may indicate that the questions are not discriminating well in our study setting [23]. The questions concern the mode of transmission, the type of sex act, and a change in a sex partner, which are all uncommon in our nations, since culture, law, and religion in the Arab world all support normal sexual relations with a single partner.

In our study, 94.8 percent of participants correctly answered question 14, indicating that the majority of participants believed HIV/AIDS was closely linked to sexual activity with more than one partner. However, more than half of the participants still believe that deep kissing with an HIV-positive partner does not transmit HIV and that using Vaseline or baby oil with condoms can reduce the risk of contracting HIV even though deep kissing and Vaseline oil do transmit the virus. Our study has both strengths and drawbacks. Over a thousand people from five different Arab countries took part in this first-of-its-kind study to adapt and validate HIV-KQ-18 for use in the Arab community. The first drawback, the study was conducted using an online method, which may have made the measure inaccessible to some individuals and susceptible to self-selection bias. Second, participants were only recruited from university students with Internet access. Therefore, they did not adequately represent the entire population. Future studies involving more diverse samples of all ages and literacy levels would aid in establishing the validity and utility of the HIV-18 KQ Arabic version. Thirdly, test-retest and inter-rater reliability, as well as the sensitivity and specificity of the scale, were not investigated.

## Conclusion

The HIV-18 KQ is a widely used tool with numerous translations around the world. The Arabic version of the HIV-18 KQ is a valid and reliable tool for assessing HIV knowledge. We

hope that our study will inspire researchers to conduct additional studies in all Arabic-speaking countries for all age groups and literacy levels.

## Supporting information

**S1 Appendix. Arabic version of the HIV knowledge questionnaire (HIV-Kq-18).**
(DOCX)

## Acknowledgments

I'd like to thank the linguists that translate the tool both forward and backward (Lina Muneer, Hussien Faied, Nour Fakih, Ahmed Elrashed).

Alpha Research Organization (ARO) team of collaborators: Rais Mohammed Amir, Judy Bassiouny, Emad Addin Zawaneh, Asmaa Mohamed Abbas, Laith Shakhatreh, Enas Elshabrawy, Buthaina Ameen.

## Author Contributions

**Conceptualization:** Mohamed Terra, Mohamed Baklola, Dina Gamal Shaheen, Abdel-Hady El-Gilany.

**Data curation:** Mohamed Terra, Mohamed Baklola.

**Formal analysis:** Mohamed Terra, Mohamed Baklola, Elfatih A. Hasabo.

**Funding acquisition:** Mohamed Terra, Mohamed Baklola.

**Investigation:** Mohamed Terra, Mohamed Baklola, Elfatih A. Hasabo.

**Methodology:** Mohamed Terra, Mohamed Baklola, Elfatih A. Hasabo.

**Project administration:** Mohamed Terra, Mohamed Baklola.

**Resources:** Mohamed Terra, Mohamed Baklola.

**Software:** Mohamed Terra, Mohamed Baklola.

**Supervision:** Mohamed Terra, Dina Gamal Shaheen, Abdel-Hady El-Gilany.

**Validation:** Mohamed Terra, Mohamed Baklola, Elfatih A. Hasabo.

**Visualization:** Mohamed Terra, Mohamed Baklola, Elfatih A. Hasabo.

**Writing – original draft:** Mohamed Terra, Mohamed Baklola, Elfatih A. Hasabo.

**Writing – review & editing:** Mohamed Terra, Mohamed Baklola, Elfatih A. Hasabo, Dina Gamal Shaheen, Abdel-Hady El-Gilany.

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
