## [Decision Letter · Decision Letter 0]

27 Feb 2023

PONE-D-23-01822Translation, Validation and cultural adaptation of the Arabic version of the HIV knowledge questionnaire (HIV-Kq-18)PLOS ONE

Dear Dr. Hasabo,

Thank you for submitting your manuscript to PLOS ONE. After careful consideration, we feel that it has merit but does not fully meet PLOS ONE’s publication criteria as it currently stands. Therefore, we invite you to submit a revised version of the manuscript that addresses the points raised during the review process.

We look forward to receiving your revised manuscript.

Kind regards,

Nour Amin Elsahoryi, pHD

Academic Editor

PLOS ONE

Journal Requirements:

4. One of the noted authors is a group or consortium: Alpha Research Organization (ARO) team of collaborators 

In addition to naming the author group, please list the individual authors and affiliations within this group in the acknowledgments section of your manuscript. Please also indicate clearly a lead author for this group along with a contact email address.

6. Please upload a copy of Figure 1 and 2, to which you refer in your text on page 8 and 10. If the figure is no longer to be included as part of the submission please remove all reference to it within the text.

7. Please ensure that you refer to Figure 3 and 4 in your text as, if accepted, production will need this reference to link the reader to the figure.

Reviewers' comments:

Reviewer's Responses to Questions

**Comments to the Author**

1. Is the manuscript technically sound, and do the data support the conclusions?

Reviewer #1: Yes

Reviewer #2: Yes

2. Has the statistical analysis been performed appropriately and rigorously? 

Reviewer #1: Yes

Reviewer #2: Yes

3. Have the authors made all data underlying the findings in their manuscript fully available?

Reviewer #1: Yes

Reviewer #2: No

4. Is the manuscript presented in an intelligible fashion and written in standard English?

Reviewer #1: Yes

Reviewer #2: No

5. Review Comments to the Author

Reviewer #1: The manuscript ‘Validation and cultural adaptation of the Arabic version of the HIV knowledge questionnaire (HIV-Kq-18)’ addresses an interesting problem in the use of HIV-18 KQ. Although there are some points and concepts need to be fixed in order to be acceptable for publishing (see the attachment).

Reviewer #2: The paper titled “Translation, Validation and cultural adaptation of the Arabic version of the HIV knowledge questionnaire (HIV-Kq-18)” seemed a statistically solid paper, however more discussion on the problem background and the tools applied are warranted.

Below are some comments to the authors:

• In the abstract background and on line 88 in the introduction, please provide the actual number of cases in addition to the percentage provided and also specify the time period.

• In the abstract, please, mention the validity metrics used to assess the questionnaire validity.

• Please, add reference for lines 121-122.

• A typo on line 132.

6. PLOS authors have the option to publish the peer review history of their article (what does this mean?). If published, this will include your full peer review and any attached files.

Reviewer #1: No

Reviewer #2: No

While revising your submission, please upload your figure files to the Preflight Analysis and Conversion Engine (PACE) digital diagnostic tool, https://pacev2.apexcovantage.com/. PACE helps ensure that figures meet PLOS requirements. To use PACE, you must first register as a user. Registration is free. Then, login and navigate to the UPLOAD tab, where you will find detailed instructions on how to use the tool. If you encounter any issues or have any questions when using PACE, please email PLOS at figures@plos.org. Please note that Supporting Information files do not need this step.<quillbot-extension-portal></quillbot-extension-portal>

---

## [Author Response · Author response to Decision Letter 0]

14 Mar 2023

Dear reviewers,

The following are the responses to your comments.

Respond to reviewer 1

We would like to thank reviewer 1 for all his effort during reviewing our manuscript.

Specific Responses: 

Comments:

• Line 118: “implying that a 10:1 ratio” must be changed by “implying that a 10.1 ratio”

• Line 174: “RStudio Version” must be changed by “R Studio Version”

• Line 174: “nFactors” must be changed by “n Factors”

• Line 175: “validateR” must be changed by “validate R”

• In page 19 the figure named “Adaptation and Validation of HIV-KQ-18 for Arabic Population” is Figure 1 not 3.

• In page 20 the figure named “The non-graphical solution of Scree plot to determine the number of factors to be retained” is Figure 2 not 4.

• In page 21 the table named “Participants’ characteristics” is Table 1 not 3.

• In page 18 the “Table 2: Factor analysis of HIV-KQ-18 Arabic Version” must be Table 4: Factor analysis of HIV-KQ-18 Arabic Version.”

Reply: The required amendments were done. Also, we uploaded Arabic version of the HIV knowledge questionnaire as a supplementary appendix.

Respond to reviewer 2

We would like to thank reviewer 2 for all his effort during reviewing our manuscript.

Specific Responses: 

Comment 1: “

• In the abstract and background on line 88 in the introduction, please provide the actual number of cases in addition to the percentage provided and also specify the time period.

• In the abstract, please, mention the validity metrics used to assess the questionnaire validity.

• Please, add reference for lines 121-122.

• A typo on line 132.”

Reply: 

• We clarified this in the introduction section by adding a paragraph from line 87 – 94 and made it as requested.

• We mentioned the validity metrics from line 59 to 63 in the methods section of the abstract.

• We added a reference to lines 121 – 122. 

• We rephrase the sentence on line 132. 

• Also, we uploaded Arabic version of the HIV knowledge questionnaire as a supplementary appendix.

Sincerely,

Elfatih A. Hasabo

---

## [Editor Report · Decision Letter 1]

3 Apr 2023

Translation, Validation and cultural adaptation of the Arabic version of the HIV knowledge questionnaire (HIV-Kq-18)

PONE-D-23-01822R1

Dear Dr, 

We’re pleased to inform you that your manuscript has been judged scientifically suitable for publication and will be formally accepted for publication once it meets all outstanding technical requirements.

Kind regards,

Nour Amin Elsahoryi, pHD

Academic Editor

PLOS ONE

<quillbot-extension-portal></quillbot-extension-portal>

---

## [Editor Report · Acceptance letter]

5 Apr 2023

PONE-D-23-01822R1 

Translation, Validation and cultural adaptation of the Arabic version of the HIV knowledge questionnaire (HIV-Kq-18) 

Dear Dr. Hasabo:

I'm pleased to inform you that your manuscript has been deemed suitable for publication in PLOS ONE. Congratulations! Your manuscript is now with our production department. 

Kind regards, 

on behalf of

Dr. Nour Amin Elsahoryi 

Academic Editor

PLOS ONE